# Disruption of Multiple Overlapping Functions Following Stepwise Inactivation of the Extended Myc Network

**DOI:** 10.3390/cells11244087

**Published:** 2022-12-16

**Authors:** Huabo Wang, Taylor Stevens, Jie Lu, Merlin Airik, Rannar Airik, Edward V. Prochownik

**Affiliations:** 1Division of Hematology/Oncology, UPMC Children’s Hospital of Pittsburgh, Pittsburgh, PA 15224, USA; 2Division of Nephrology, UPMC Children’s Hospital of Pittsburgh, Pittsburgh, PA 15224, USA; 3Department of Developmental Biology, The University of Pittsburgh Medical Center, Pittsburgh, PA 15224, USA; 4The Department of Microbiology and Molecular Genetics, The University of Pittsburgh Medical Center, Pittsburgh, PA 15261, USA; 5The UPMC Hillman Comprehensive Cancer Center, Pittsburgh, PA 25232, USA; 6Pittsburgh Liver Research Center, University of Pittsburgh, Pittsburgh, PA 15261, USA

**Keywords:** aging, ChREBP, DNA repair, Mga, MondoA, Mnt, Mxd, c-Myc, N-Myc, senescence, telomerase

## Abstract

Myc, a member of the “Myc Network” of bHLH-ZIP transcription factors, supervises proliferation, metabolism, and translation. It also engages in crosstalk with the related “Mlx Network” to co-regulate overlapping genes and functions. We investigated the consequences of stepwise conditional inactivation of Myc and Mlx in primary and SV40 T-antigen-immortalized murine embryonic fibroblasts (MEFs). *Myc*-knockout (*Myc*KO) and *Myc* × *Mlx* “double KO” (DKO)—but not *Mlx*KO—primary MEFs showed rapid growth arrest and displayed features of accelerated aging and senescence. However, DKO MEFs soon resumed proliferating, indicating that durable growth arrest requires an intact Mlx network. All three KO MEF groups deregulated multiple genes and functions pertaining to aging, senescence, and DNA damage recognition/repair. Immortalized KO MEFs proliferated in Myc’s absence while demonstrating variable degrees of widespread genomic instability and sensitivity to genotoxic agents. Finally, compared to primary *Myc*KO MEFs, DKO MEFs selectively downregulated numerous gene sets associated with the p53 and retinoblastoma (Rb) pathways and G_2_/M arrest. Thus, the reversal of primary *Myc*KO MEF growth arrest by either Mlx loss or SV40 T-antigen immortalization appears to involve inactivation of the p53 and/or Rb pathways.

## 1. Introduction

The c-Myc (Myc) oncoprotein is a critical bHLH-ZIP transcription factor that regulates hundreds–thousands of target genes in association with its bHLH-ZIP partner protein Max [1,2]. Positive control is achieved via the direct binding of Myc–Max heterodimers to consensus E-box elements (CACGTG) that are usually located in proximity to transcriptional start sites [2,3]. This allows Myc to recruit an assortment of transcriptional cofactors and chromatin modifiers that coordinately license RNA Pol II activation, relieve transcriptional pausing, and promote transcriptional readthrough [4]. Negative transcriptional regulation by Myc is more indirect and mediated via the direct interaction of Myc or Myc–Max heterodimers with certain positively acting transcription factors such as Sp1, Sp3, and Miz1/ZBTB17, in ways that inhibit binding to their own consensus sites-namely, Sp1 and InR elements-and/or quench their transcriptional activity at these sites [5,6]. Myc is normally expressed during log-phase growth, is suppressed in quiescent and differentiated cells, and is tightly controlled at multiple levels. In cancer, however, *MYC* gene amplification/translocation or aberrant growth factor signaling can cause accumulation of supraphysiological levels of Myc protein [4,7]. Alternatively, Myc may be stabilized due to mutations in factors that promote its degradation or, less commonly, by point mutations in Myc mRNA or the protein itself [7]. Regardless of the underlying cause, excessive levels of Myc can dysregulate its normal target genes, as well as those with otherwise low-affinity E-boxes (i.e., “pathologic” targets), which then contribute to neoplastic initiation or progression [2,3,8].

The attenuation of Myc expression that occurs with the onset of cellular quiescence and/or differentiation is accompanied by a reversal of the abovementioned chromatin modifications and transcriptional silencing of its positively regulated direct target genes [4]. This too is an active process mediated by six additional bHLH-ZIP factors whose expression is both tissue-specific and developmentally determined [2]. Heterodimers between Max and these so-called “Mxd members” (i.e., Mxd1-4, Mnt, and Mga) compete for many of the sites occupied by Myc while also recognizing unique sites [2,3]. In the former case, these displace Myc-Max heterodimers and reverse the Myc-directed epigenetic modifications that accompany transcriptional activation [2,4]. The carefully orchestrated and highly dynamic balance between Myc and Mxd factors, all in association with their common partner Max, dictates the transcriptional output of this highly integrated “Myc Network”. Prominent categories of genes under its supervision include those whose products contribute to mitochondrial and ribosomal structure and function, cell-cycle control and extramitochondrial metabolism-notably, glycolysis and glutaminolysis [2,8,9,10,11].

The more recently characterized “Mlx Network” is also composed of bHLH-ZIP transcription factors that are structurally and functionally related to Myc network members [2,12]. It directly regulates a smaller and less functionally diverse repertoire of direct target genes than does the Myc network, although considerable overlap has been identified [2,3,11,12]. Two Myc-like factors—ChREBP and MondoA, each with its own distinct tissue-specific and developmentally regulated expression patterns—heterodimerize with the Max-like factor Mlx [2,12]. Unlike Myc, which is a nuclear protein, the subcellular distribution of ChREBP and MondoA is reliant upon certain metabolites—notably, glucose, glucose-6-phosphate, lactate, and adenine nucleotides [2,12]. The binding of these small molecules allows ChREBP-Mlx and MondoA-Mlx heterodimers to translocate to the nucleus and associate with “carbohydrate response elements” (ChoREs), which are classically composed of two E-boxes separated by five nucleotides, although substantial cross-binding to pure E-boxes also occurs [1,2,3]. The association of Mxd1, Mxd4, and Mnt with either Max or Mlx also affords an additional means of direct crosstalk between the Myc and Mlx networks, collectively referred to as the “Extended Myc Network” [2,3,12]. We have proposed that the non-random close proximity of E-boxes and ChoREs in the promoters of some genes allows for combinatorial binding of these different heterodimers±, their direct interaction, and the fine-tuning of their transcriptional output, while allowing for both transitory and variable degrees of metabolite responsiveness that are precisely matched with proliferative and environmental cues [2,3].

Body-wide *Myc* inactivation in mice is embryonic lethal, whereas in individual tissues the consequences of its loss are tissue-, age-, and context-dependent [13,14]. In early-passage primary murine embryonic fibroblasts (MEFs), for example, incremental reductions in Myc cause progressive and eventually complete cell-cycle arrest, whereas in one highly studied line of immortalized rat fibroblasts, total *Myc* gene knockout (KO) leads to a less pronounced proliferative slowing in association with severe mitochondrial impairment [10,15,16]. Some primary adrenal pheochromocytomas and paraganglioneuromas, along with cell lines derived from them, show *MAX* gene inactivation and, thus, at least partial Myc independence [2]. The in vivo expansion of murine pancreatic β-cells in response to hyperglycemia is also Myc-dependent, although this diminishes with age [17]. *Myc* KO in hepatocytes affects neither their long-term in vivo replication nor their susceptibility to transformation, but it does impair the growth of experimental hepatoblastomas, whereas *Chrebp* KO slows the proliferation of both non-transformed and transformed hepatocytes [11,14]. Finally, global *Mlx* KO, which functionally inactivates both ChREBP and MondoA, does not affect normal embryonic development but does cause male infertility and apoptosis of germ cells, whereas *Mlx* KO in the liver inhibits long-term hepatocyte regeneration and proliferation of hepatoblastomas even more than *Chrebp* KO, although it is permissive for the development of benign hepatic adenomas [1,2,3]. Collectively, these findings underscore the variable, tissue-specific, and developmentally dependent consequences associated with the selective elimination of extended Myc network members.

To explore the nature and consequences of the crosstalk between the Myc and Mlx networks in greater detail, in this paper we report the immediate and longer-term ramifications of *Myc* and/or *Mlx* KO in both primary and immortalized MEFs under defined in vitro conditions. This “step-wise” inactivation, which progresses from the KO of *Myc* to the KO of *Mlx*, and then to the dual KO of both factors, permits us to demonstrate definitively that each network differentially impacts many normal functions in ways that correlate with both their shared and unique target gene expression profiles. It also identifies new roles for the Myc and Mlx networks in normal aging and senescence, the maintenance of chromosomal stability, DNA damage recognition and repair, and the support of Myc-independent growth. Our findings reinforce the notion that each arm of the extended Myc network possesses distinct capabilities while cooperating in complex ways to ensure that they are properly balanced.

## 2. Materials and Methods

### 2.1. Mice, MEFs, and Documentation of Target Gene Excision Efficiency

All breeding, care, husbandry, and procedures were approved by the University of Pittsburgh’s Department of Laboratory and Animal Resources (DLAR) and the Institutional Animal Care and Use Committee (IACUC), with standard animal chow and water provided ad libitum. Mouse strains containing two “floxed” alleles of the *Myc* and/or *Mlx* genes and described previously [2] were crossed with a strain carrying two copies of the ROSA26-CreER transgene (B6.129-*Gt (ROSA)26Sor^tm1(cre/ERT2)Tyj^*/J) (The Jackson Laboratory, Inc., Bar Harbor, ME, USA), which expresses a Cre recombinase-estrogen receptor (CreER) fusion transgene driven by the ubiquitously expressed ROSA26 promoter. Homozygosity at all alleles was confirmed by quantitative TaqMan assays.

Standard MEF cultures were generated from ~e14–16 embryos obtained from 2–3 pregnant dams from each group, with the cells from a total of 8–12 embryos per group being pooled and frozen at −80 °C within six days and two passages of in vitro establishment. All MEFs were maintained in Dulbecco’s modified minimum essential medium (DMEM) containing 10% fetal bovine serum, glutamine, and penicillin/streptomycin [10]. To excise the *Myc* and/or *Mlx* loci, MEFs were re-established from frozen stocks and allowed to attain log-phase growth before being exposed to 500 nM 4-hydroxytamoxifen (4OHT) or the 100% ethanol vehicle alone, which were changed daily. At various points throughout the treatment, three replica wells of cells were harvested by trypsinization, washed twice in PBS, and frozen at −80 °C. DNAs were then extracted using a Qiagen DNeasy kit (Qiagen, Inc., Germantown, MD, USA) and used in a quantitative TaqMan-based assay to obtain the relative ratios of intact and excised alleles for each target gene, as previously described (Appendix A) [3].

All cell counts were performed on trypsinized cultures using a Vi-CELL cell viability analyzer (Beckman-Coulter, Inc., Pasadena, CA, USA).

### 2.2. MEF Immortalization

A lentiviral vector encoding SV40 T-antigen and blasticidin resistance (GenTarget, Inc., San Diego, CA, USA) was used to transduce each of the three abovementioned MEF cell lines in 12-well plates. Following 72 h of exposure to the viral stock (MOI = 1), the cells were expanded over the next week, at which point blasticidin (Thermo Fisher, Inc., Pittsburgh, PA, USA) was added at a concentration of 1 μg/mL for 5 days and then 2 μg/mL for an additional 10 days. Under these conditions, non-transduced cells were completely killed by the antibiotic. Quantification of target gene excision was then performed as described above for primary MEFs, except that the length of 4OHT exposure was extended to 10 days. The excision of each gene was confirmed by a combination of qPCR and Western blotting for protein expression (Appendix A).

### 2.3. Puromycin Labeling, Immunoblotting, and Staining

Puromycin labeling was performed on cells that were seeded so as to achieve ~80% confluence by the following day. The drug was added to standard DMEM + 10% FCS at a final concentration of 10 μg/mL, and cells were incubated for 1 h at 37 ℃. For these and all other all immunoblotting experiments, cells were harvested, lysed, and processed as previously described [10,11]. All antibodies used, their dilutions, and the vendors from which they were obtained are shown in Appendix A. All immunoblots were developed using a Pierce Enhanced ECL chemiluminescence detection kit (Thermo Fisher, Inc., Pittsburgh, PA, USA) [10,11].

### 2.4. Live Cell and Immunohistochemical Staining, Imaging, and Flow Cytometry

For phase-contrast microscopy, live cells were imaged with a Revolve microscope model RVL-100-G (Echo Laboratories, Inc., San Diego, CA, USA). Other cells were plated under the conditions described above and were stained with Hoechst 33324, MitoTracker™ Green (both from Life Technologies, Inc., Carlsbad, CA, USA), and Cell Mask™ Orange Actin Tracking Stain (Thermo-Fisher, Inc., Pittsburgh, PA, USA) using the directions provided by the suppliers (Appendix A). Imaging was performed on a STELLARIS 5 Confocal Microscope (Leica Microsystems, Wetzlar, Germany). For immunohistochemical (IHC) staining, cells were plated onto glass coverslips and allowed to attach overnight, at which point they were then fixed in 4% paraformaldehyde-PBS, stained with hematoxylin-eosin (H & E) or the indicated antibodies (Appendix A), and imaged with the same microscope. TdT assays were performed using a TUNEL Assay Kit (ab206386) according to the directions of the vendor (Abcam, Inc., Cambridge, England). Cells exposed to etoposide were plated in the same manner but exposed to the drug on the following day. Fluorescence intensities and foci numbers in IHF samples were quantified using ImageJ software (https://imagej.nih.gov/ (accessed on 1 June 2022)) and graphed with the R package ggplot2.

Propidium iodide staining to assess the cell cycle was performed in isolated nuclei as previously described [18]. Reactive oxygen species were measured by exposing monolayer cultures of MEFs maintained at 37 ℃ to fresh DMEM containing CM-H_2_DCFDA or MitoSOX™ Red. Mitochondrial mass was assessed by staining with acridine orange 10-nonyl bromide (NAO: Thermo-Fisher), and neutral lipid content was quantified by staining with BODIPY™ 505/515 (Life Technologies, Inc.) [2,14]. Staining with the β-galactosidase substrate CellEvent^TM^ Senescence Green and the fluorescent glucose analog 2-(*N*-(7-nitrobenz-2-oxa-1,3-diazol-4-yl)amino)-2-deoxyglucose (2-NBDG) (both from Thermo Fisher) was performed according to the directions of the suppliers (Appendix A). All quantifications were performed on 6 biological replicates comprising 20,000 cells/sample using a FlowJo v10 flow cytometer (FlowJo LLC, Ashland, OR, USA) (Becton-Dickinson Biosciences, San Jose, CA, USA). Cell-cycle analyses were performed using ModFit LT Version 3.3.11 (Verity Software House, Topsham, ME, USA), as previously described [18].

### 2.5. Lactate Measurements

~Day 10 primary MEFs were plated so as to allow replicative arrest of WT and *Mlx*KO cells to be achieved over the next 1–2 days. *Myc*KO and DKO cells were plated at higher densities to account for their growth-arrested state. The cells were then maintained for three additional days in a medium containing 0.2% FCS to further consolidate quiescence, at which point lactate levels were quantified and normalized to total protein [19]. The medium from quiescent MEFs was removed and, where necessary, it was diluted in fresh medium to allow lactate concentrations to be within the linear range of the test strips used for detection (Lactate Plus Analyzer, Sports Resource Group, Inc., Hawthorne, NY, USA). The results were calculated from standard curves.

### 2.6. Respirometry Studies

Mitochondrial oxygen consumption was quantified as previously described using an Oroboros Oxygraph 2k respirometer (Oroboros Instruments, Innsbruck, Austria) [19]. Briefly, WT or KO cells previously treated with 4OHT were plated into 100 mm tissue culture plates and allowed to attach for 24 h. After harvesting, 3 × 10^6^ cells in MiR05 buffer containing cytochrome c (final concentration 10 μM) (Oroboros) were added to the oxygen consumption measurement chamber and permeabilized by the addition of digitonin (final concentration 16.6 μM). Oxygen consumption rates (OCRs) were then quantified during the sequential addition of the following TCA cycle substrates or inhibitors: malate (final concentration 2.0 mM), pyruvate (final concentration 5 mM), succinate (final concentration 10 mM), glutamate (final concentration 10 mM), and rotenone (final concentration 0.5 μM). All substrates were purchased from Sigma-Aldrich (St. Louis, MO, USA). The results were then adjusted to account for any differences in the initial total protein levels.

### 2.7. Transcriptional Profiling and Bioinformatics Analyses

MEF RNAs were isolated using RNeasy kits, according to the directions of the supplier (Qiagen, Inc., Germantown, MD, USA). RIN values were determined with an Agilent 2100 Bioanalyzer (Agilent Technologies, Foster City, CA, USA), and all samples had RIN values ≥ 9.6. Poly-A-enriched sequencing libraries were generated from triplicate sets of RNAs from control WT MEFs and each KO group using a NEBNext Ultra Directional RNA Library Prep kit (New England Biolabs, Beverly, MA, USA), as previously described [3]. Sequencing was performed on a NovaSeq 6000 instrument (Illumina, Inc., San Diego, CA, USA) produced by Novagene, Inc. (Sacramento, CA, USA) [3]. Original raw data are available through Gene Expression Omnibus (GEO) access GSE210362. CLC Genomic Workbench version 22 (Qiagen) was used to map raw reads to the GRCm39 mouse reference genome. To identify common and unique transcripts among the KO cell lines relative to WT cells, we used three different computational methods (DESeq, CLC Genomics Workbench, and edgeR) [20]. Only those transcripts found to be differentially expressed by all three methods and with *q* values < 0.05 relative to WT cells were included in this survey, providing the most conservative estimate of transcriptional profiles.

For gene set enrichment analysis (GSEA), the Enrichr and MSigDB databases (http://amp.pharm.mssm.edu/Enrichr and https://www.gsea-msigdb.org/gsea/ (accessed on 1 June 2022)) were screened using clusterProfiler (R package version 4.2) (https://bioconductor.org/packages/release/bioc/html/clusterProfiler (accessed on 1 June 2022)). Differentially regulated gene sets were then displayed using the ridgeline plot application tool ridgeplot, the hierarchical clustering tool treeplot, and the GSEA result visualization tool gseaplot [21,22]. To identify correlations between Myc and Mlx transcript levels in human primary adult and pediatric cancers, we accessed the GDC Pan-Cancer (PANCAN) database (https://xenabrowser.net/datapages/?cohort=GDC%20Pan-Cancer%20(PANCAN)&removeHub=https%3A%2F%2Fxena.treehouse.gi.ucsc.edu%3A443 (accessed on 1 June 2022)) and downloaded transcriptomic profiles from the Cancer Genome Atlas (TCGA) and Therapeutically Applicable Research To Generate Effective Treatments (TARGET) (https://ocg.cancer.gov/programs/target (accessed on 1 June 2022)) cancer sets. Genome-wide binding sites for Myc and Mlx were identified as previously described [3]. Briefly, we down-loaded ChIP-Seq data from 22 experiments performed with 12 human and murine cell lines from the ENCODE website (Appendix A) [23]. We used ChIPpeakAnno version 3.13 and the annotation R packages “TxDb.Hsapiens.UCSC.hg38.knownGene” or “TxDb.Mmusculus.UCSC.mm10.knownGene” to annotate Myc and Mlx ChIP-Seq peaks to ± 2.5 kb of the transcriptional start sites of all protein-coding genes (Appendix A) [24].

Statistical Analyses were preformed using R software v4.1.0 (R Foundation for Statistical Computing, Vienna, Austria) and GraphPad Prism v9.00 (GraphPad Software Inc., New York, NY, USA) [3,14].

## 3. Results

### 3.1. The Morphology and Proliferation of Primary MEFs Are More Dependent on Myc Than on Mlx

Primary MEFs were established from ~e14–16 embryos bearing two “floxed” alleles of the *Myc* and/or *Mlx* genes and two copies of a ROSA26 promoter-driven transgene encoding a Cre recombinase-estrogen receptor (CreER) fusion protein (Appendix A) [3]. Following an initial 2–3 day expansion, the cells were re-plated and maintained for at least 7 days in medium containing 500 nM 4-hydroxytamoxifen (4OHT). This achieved > 95% excision efficiency of each target gene along with comparable reductions in protein expression (Appendix A). We refer hereafter to MEFs with intact *Myc* and/or *Mlx* loci as wild-type (WT), and to those with excised loci as *Myc*KO, *Mlx*KO, and double-knockout (DKO) cells (and collectively as KO cells). Following excision of each target gene, KO cells were routinely maintained in medium containing 250 nM 4OHT, which was removed for at least 3 days prior to performing any subsequent studies. Because all three WT MEF lines behaved similarly in all experiments, results with only a single such line were used as controls in most cases.

Other than occasional, somewhat larger-appearing *Myc*KO and DKO cells, few morphological differences between the MEF groups were noted on day 7 (not shown). By ~day 14, however, most *Myc*KO and DKO cells had become noticeably larger-appearing, flatter, and less spindle-shaped, while also showing the high cytoplasmic:nuclear ratios associated with aged and senescent primary fibroblasts and immortalized *Myc*^−/−^ rat fibroblasts (Figure 1A,B) [10,14]. Confocal microscopy confirmed these findings, in addition to demonstrating more numerous and elongated actin filaments in *Myc*KO and DKO MEFs and an apparent increase in mitochondria (Figure 1C,D). Consistent with previous studies on MEFs expressing low–undetectable Myc levels from the time of conception, *Myc*KO and DKO MEFs showed little tendency to proliferate following their re-plating on day 7 (Figure 1E–G) [16]. These results further established that acute Myc loss in primary or established cell lines rapidly alters their initial proliferative status and, somewhat more belatedly, their morphology [10,15,16,25]. Unlike previous results in hepatocytes and germ cells, in which *Mlx* inactivation dramatically slows in vivo proliferation, it had little effect on the morphology, survival, or growth of primary MEFs [1,3]. Therefore, in primary cells, the consequences of *Myc* and *Mlx* inactivation are highly context- and/or tissue-specific.

In a final control experiment, WT MEFs bearing floxed Myc alleles were transduced with a retroviral vector encoding a MycER fusion protein and puromycin resistance. After selecting stably transduced pooled clones in a puromycin-containing medium, we simultaneously excised the endogenous *Myc* locus while also activating a low level of MycER expression by maintaining cells in 60 nM 4OHT. After 10 days of 4OHT exposure, these cells were re-plated while maintaining them in 4OHT, and their growth rates were compared to those of similarly treated cells that did not express MycER. As shown in Figure 1E, the latter cells failed to proliferate over the ensuing 4 days, whereas the former increased by 4.2 ± 0.4-fold (*p* < 0.01). Consistent with the idea that these *Myc*KO cells had been rescued by conditionally re-expressing a low level of Myc, they also maintained the overall morphology of the WT cells (Figure 1A,B, not shown).

### 3.2. Loss of Myc and/or Mlx Differentially Affects Cell-Cycle Progression and Mitochondrial Structure and Function

The genetic or pharmacological inhibition of Myc in normal cells is accompanied by a rapid proliferative arrest in G_o_/G_1_, whereas in transformed cells this may also occur in S and/or G_2_/M, depending on the intactness of other cell-cycle checkpoints (e.g., 10, 16, 18). Consistent with previous findings, growth-arrested day 7 *Myc*KO MEFs contained noticeably larger G_o_/G_1_ and smaller S-phase populations relative to proliferating WT cells (Figure 2A,B). In keeping with their nearly normal proliferative rate, day 7 *Mlx*KO cells showed a much more modest reduction in their S-phase population, a larger G_2_/M population, and virtually no change in their G_o_/G_1_ population. DKO cells showed intermediate profiles that more closely resembled those of *Myc*KO cells and mirrored their similarly growth-arrested state. These findings confirmed that *Myc* loss in primary MEFs inhibited cell-cycle progression in association with a selective G_o_/G_1_ arrest and continued to do so in early-passage DKO MEFs. The less pronounced—but still significant—cell-cycle changes associated with *Mlx*KO MEFs did not appreciably alter their proliferation as happens in primary hepatocytes [3]. Finally, the combined KO of *Myc* and *Mlx* in DKO cells altered cell-cycle profiles in ways that were distinct from those seen following KO of either individual gene, indicating that the Myc and Mlx networks influence one another’s functions.

By day 14, further consolidation of the above cell-cycle profiles was noted, which included the accumulation of even smaller S-phase populations in *Myc*KO and DKO cells and an *Mlx*KO cell population that more closely resembled that of day 7 WT cells (Figure 2A,B). Thus, beyond day 7, the loss of *Myc* and/or *Mlx* continued to alter the cell-cycle status for as long as an additional week. The prominent G_0_/G_1_ population, previously seen in day 7 *Myc*KO MEFs, was reduced in day 14 MEFs and was replaced by a significantly larger G_2_/M population. Day 7 cells thus continued a slow transition from S-phase into a more durable arrest in G_2_/M, without progressing through mitosis. Together, these findings demonstrated that, in addition to the impact of *Myc* and *Mlx* loss on the cell cycle being distinct and cooperative, it occurred gradually, extended well beyond the point of excision of the intended target genes, and modified the consequences of both. The numerous other significant but minor differences between the KO cell lines (Figure 2B) were further testimony to each factor’s specific roles in maintaining cell-cycle control.

The compromise of Myc and/or Mlx causes tissue-specific defects in oxidative metabolism that may include—but are not necessarily limited to—the loss of mitochondrial mass, reduced TCA cycle substrate oxidation, and an overreliance on fatty acid β-oxidation as an energy source, which is associated with a larger-than-needed uptake of fatty acids and storage of the excess in the form of neutral lipids [3,10,11,14,25]. Some of these properties are recapitulated in the livers of mice with compromised Myc and/or Mlx networks, who develop progressive steatosis that reflects this unbalanced substrate utilization [3,11,14]. As a consequence of electron transport chain (ETC) dysfunction, *Myc*KO cells also generate more reactive oxygen species (ROS), which promote further pathological changes due to lipid peroxidation and oxidative DNA damage [10,14].

Unlike the mitochondrial atrophy and neutral lipid accumulation that accompany Myc inactivation in immortalized *Myc^−/−^* rat fibroblasts [10], day 7 and day 14 *Myc*KO and DKO MEFs actually increased their mitochondrial mass by 1.6-2-fold, while still accumulating neutral lipids (Figure 2C,D). *Mlx*KO MEFs on the other hand, initially mimicked the mitochondrial mass gain of *Myc*KO and DKO MEFs but, by day 14, showed a small reduction. At neither time was there any change in the neutral lipid content. The findings in *Myc*KO and DKO MEFs were consistent with the perceived increase in mitochondrial content observed by confocal microscopy, as well as with previously documented increases in mitochondrial mass and dysfunction that accompany senescence (Figure 1C,D) [26]. Further consistent with the differences in observed mitochondrial mass, ROS production—much if not all of which was of mitochondrial origin—was increased in *Myc*KO and DKO MEFs (Figure 2E,F). Together, these findings indicated that *Myc* KO was associated with variable degrees of aberrant mitochondrial structure and function that only partially recapitulated those of immortalized *Myc^−/−^* rat fibroblasts [10,15].

*Myc*KO and *Mlx*KO primary hepatocytes, *Myc^−/−^* rat fibroblasts, and other normal and tumor cells that over- or underexpress *Myc* or *Mlx* display variable degrees of aberrant oxidative phosphorylation (Oxphos) [3,8,10,11,25,27]. In contrast, minimal differences in baseline oxygen consumption rates (OCRs), responses to different TCA cycle substrates, and the functions of ETC complexes I or II were noted among the abovementioned MEF cell lines (Figure 2G). However, given the greater mitochondrial mass of *Myc*KO and DKO cells relative to WT cells and their overall inefficiency in electron transfer (Figure 2C,E,F), the simplest interpretation of these results is that the seeming equivalency of OCRs among the four MEF groups was achieved at the expense of increasing the total mass of functionally inefficient mitochondria in *Myc*KO and DKO cells.

### 3.3. MycKO and DKO MEFs Display Features of Premature Aging and Senescence

The altered morphology, mitochondrial mass, Oxphos, and ROS production of *Myc*KO and DKO MEFs also accompany normal and premature senescence and aging [26]. Primary cells enter a state of irreversible senescence as a result of prolonged in vitro propagation or become prematurely senescent in response to stresses such as oncogene activation, suppression of DNA repair pathways, or genotoxic insults, which in some cases can be mitigated by Myc overexpression [28,29,30]. Such cells may display additional—albeit variable—markers of senescence and/or premature aging, such as altered size and lysosomal content, enhanced glucose uptake, and the increased expression of senescence-associated β-galactosidase (SA-β-gal), while decreasing their rates of protein synthesis [27,31,32,33]. The appearance of these features, their relative prominence, and their duration vary in response to different stimuli, among different cell types, and even among clonally derived subpopulations of cells, further underscoring the paucity of universal or even reproducible markers for the senescent state [27,31,32,33].

Guided by these considerations, we next asked whether any of the above KO MEF cell lines displayed additional evidence of premature aging and/or senescence. Increases in side-scatter populations, indicative of the increased lysosomal content of senescent cells [33], were noted in both day 7 and day 14 MycKO and DKO cells (*p* < 0.001), with little change being seen in *Mlx*KO cells (Figure 3A). Consistent with the more robust glycolytic activity of other reported senescent cell types [31], both *Myc*KO and MlxKO MEFs showed significantly higher uptake of the fluorescent glucose analog 2-NBDG relative to WT MEFs at both day 7 and day 14 (Figure 3B). In DKO cells, *Mlx* loss mitigated the more prominent 2-NBDG uptake of *Myc*KO cells. Differences in SA-β-Gal levels were also noted among day 7 and day 14 WT *MycKO* and DKO cells based on the cleavage and fluorescent activation of the SA-β-Gal substrate CellEvent™ Senescence Green (Figure 3C). The latter cells in particular showed a progressive increase in SA-β-Gal between days 7 and 14. Consistent with the lower levels of other senescence markers, *Mlx*KO MEFs showed less pronounced differences in CellEvent™ Senescence Green staining at both day 7 and day 14.

Lactate dehydrogenase A (LDHA), whose gene is a direct Myc target [34], is a critical mediator of the Warburg effect, which helps to maintain high rates of glycolysis, NAD+ generation, and cell proliferation [2,9]. The *Ldha* gene is also regulated by MondoA, although precisely how this is integrated with the Myc network is unclear [34]. In addition to having higher levels of glucose uptake, some aging and/or senescent cells also increase lactate production despite having lower rates of proliferation [35]. When plated at high density and then subjected to further proliferative arrest by reducing the serum concentration [19], WT and *Myc*KO MEFs produced equivalent amounts of lactate, despite the latter’s previously noted higher uptake of 2-NBDG (Figure 3D). In contrast, similarly maintained *Mlx*KO cells showed a pronounced increase in lactate production, while DKO cells showed intermediate levels. These results indicate that, in primary MEFs, the *Ldha* gene is subject to positive and negative regulation by the Myc and Mlx networks, respectively, with the latter’s effect tending to dominate.

Non-insulin-dependent Glut1 and insulin-dependent Glut4 are the two major high-affinity Class I glucose transporters expressed by fibroblasts, with the *Glut1* (*Slc2a1*) gene having been reported to be a direct Myc target [36]. Consistent with their higher rates of 2-NBDG uptake and lactate production, *Mlx*KO and DKO cells significantly increased their expression of Glut1, and DKO cells modestly increased their expression of Glut4 (Figure 3E). Although *Glut1* was not identified as a direct Mlx target in our screening of the ENCODE database (not shown) [23], our findings indicate that it is nonetheless negatively regulated by Mlx—perhaps indirectly.

Protein synthesis rates are reduced in aging and senescent cells, due at least in part to the impaired synthesis of ribosomal proteins, rRNAs, and tRNAs [32]. Similarly, *Myc*KO and DKO MEFs showed 60–80% reduced rates of puromycin incorporation into actively extending polypeptide chains (Figure 3F). Collectively, these and the aforementioned findings indicate that the Myc and Mlx networks, both individually and together, differentially contribute to the metabolic phenotypes associated with aging and/or senescence, some of which appear to be transient.

### 3.4. The Senescence-like State of DKO MEFs Is Spontaneously Reversible

Although *Mlx* KO alone had little if any impact on MEF proliferation (Figure 1F), it did reinforce the effects of *Myc* KO on cell-cycle arrest and the accumulation of forward- and side-scatter populations in day 14 DKO MEFs, while tempering other pro-senescent phenotypes (Figure 2A–F). These findings indicate that *Mlx* KO impacts certain features of growth and senescence in ways that are both Myc-dependent and -independent and that may either reinforce or mitigate those of *Myc* KO alone. Unexpectedly, non-proliferating DKO MEFs maintained in culture beyond days 14–21 resumed log-phase growth at rates equaling or even exceeding those of comparably passaged WT MEFs and regained a more WT-like morphology (Figure 4A,B). Forward- and side-scatter populations, 2-NBDG uptake, and CellEvent™ Senescence Green staining also normalized so as to more closely approximate those of comparably aged WT MEFs (Figure 4C–E). Measurements of mitochondrial mass, using both NAO and MitoTracker™ Green, now showed it to be significantly increased relative to WT cells, rather than decreased as had been documented for day 7 and day 14 MEFs (Figure 4F,G vs. Figure 2C). Finally, the levels of both total and mitochondria-specific ROS were equivalent (Figure 4H,I). Repeat TaqMan-based qPCR assays and immunoblots performed on several occasions verified the persistent KO of both *Myc* and *Mlx* genes and the absence of their encoded proteins (not shown). Thus, the resumption of DKO MEF proliferation between days 14 and 21 could not be attributed to a minority cell population with incompletely excised *Myc* and/or *Mlx* genes that eventually replaced the KO population. Like WT and *Mlx*KO cells, these DKO MEFs could be maintained in a continuously replicating state until ~day 100 before entering the permanent cell-cycle arrest associated with natural senescence (not shown). These findings indicate that Mlx is necessary to maintain the more durable growth-arrested state associated with *Myc*KO cells. Moreover, the resumption of growth coincided with the loss of numerous senescence markers that were expressed in earlier-passage cells.

Day >14 DKO MEFs are reminiscent of immortalized *Myc^−/−^* rat fibroblasts, which also retain the ability to proliferate (albeit very slowly), despite their atrophic mitochondria and markedly reduced ATP stores [10,15,25]. Some downstream Myc target genes can rescue these defects to variable degrees, but the true basis for the original Myc-independent growth of these cells remains enigmatic [37,38]. Hypothesizing from the aforementioned findings that altered Mlx expression might be responsible, we examined this in both *Myc^−/−^* fibroblasts and the *Myc^+/+^* parental cell line from which they were originally derived [15]. Indeed, the Mlx γ isoform, which encodes the full-length protein and is the only one of the three isoforms that can enter the nucleus [39], was markedly reduced in *Myc^−/−^* rat cells (Figure 4J). These findings were consistent with observations that some human cancers show *MLX* copy number loss, as well as with our recent documentation that Mlx is a potent suppressor of hepatic adenomatosis [2,3]. They further suggest that achieving a proper balance between the Myc and Mlx networks is necessary for regulating both tumor cell growth and cell-cycle arrest. Indeed, we identified modest but nonetheless significant positive correlations between Myc and Mlx transcript levels in 17 tumor types whose gene expression profiles were retrievable through the Cancer Genome Atlas (TCGA) and the Therapeutically Applicable Research To Generate Effective Treatments (TARGET) collections in the PANCAN database (Figure 4K). Together, these and previous findings suggest that Myc and Mlx impose different controls over proliferation and senescence in MEFs, with the former promoting growth and inhibiting senescence, and the latter exerting the opposite effects.

### 3.5. Co-Regulated Myc and Mlx Target Genes Impact Similar Pathways and Functions

RNA-Seq was performed on day 10 primary MEFs to determine the extent to which their gene expression profiles were *Myc-* and/or *Mlx-*dependent and could explain the similarities and differences in cellular properties and behaviors. To ensure high reliability and reproducibility, we employed three different methods of computational analysis (DESeq, CLC Genomics Workbench, and edgeR) and further considered only those transcripts whose differential expression relative to WT MEFs (*q* < 0.05) was verified by all three approaches [20]. In keeping with the notion that the Mlx network’s transcriptional repertoire is more limited, we first found that its 1033-member transcript set—represented by segments 2, 4, 5, and 7 shown in Figure 5A—was less than one-quarter as large as the 4387-member set of transcripts under Myc control (i.e., segments 3, 5, 6, and 7) [2,3,12]. Second, ~41–50% of the differentially expressed genes in each KO group were positively regulated. Third, only 18.3% of genes identified as Myc-regulated (803/4387) and 21.3% of those identified as Mlx-regulated (220/1033) were independent of the other factor. Conversely, 14.5% of Myc transcripts (638/4387) were co-regulated by Mlx, while 61.8% of Mlx transcripts (638/1033) were co-regulated by Myc. Finally, we relied upon data from the ENCODE consortium’s multiple cell lines to identify potential direct binding sites for Myc and Mlx residing between −2.5 and +2.5 kb of transcriptional start sites [23]. In this way, 64.1% of the 6347 unique genes represented in Figure 5A were identified as potential direct Myc targets, 26.9% as direct Mlx targets, and 22.7% as direct targets for both factors.

Not unexpectedly, transcripts represented by different segments of the Venn diagram shown in Figure 5A were subject to distinct modes of Myc and/or Mlx regulation. For example, segments 1, 2, and 3 were comprised of transcripts whose dysregulation was confined to a single cell line, whereas segment 5—under the dual control of Myc and Mlx—contained transcripts that were not dysregulated in DKO cells, and segment 7 contained transcripts that were dysregulated in all three KO cell lines (Figure 5B). Across all overlapping segments (i.e., segments 4–7), numerous genes were also identified whose changes in response to Myc and/or Mlx loss were in opposite directions. These results reinforce the idea that target genes include those that respond either positively or negatively to members of only one or both networks [2,3].

Ingenuity pathway analysis (IPA) was next used to identify the top pathways and associated functions represented by the 6348 transcripts displayed in Figure 5A, as well as to predict the directions in which these were likely to be altered relative to WT MEFs. Consistent with the profiling shown in Figure 5B, we identified functions whose predicted deregulation was restricted to only a single KO line or was shared between two or among all three KO lines (Figure 5C). The presumed activation or suppression of these functions was also not necessarily concordant. For example, “Cell Cycle Control of Chromosomal Replication” and “Kinetochore Metaphase Signaling” were predicted to be downregulated in all three KO cell lines (Figure 5C) whereas pathways involving “Cardiac Hypertrophy” and “Idiopathic Pulmonary Fibrosis Signaling” were predicted to be upregulated in *Myc*KO and DKO MEFs and downregulated or barely changed in *Mlx*KO MEFs. Elsewhere, the “Superpathway of Cholesterol Biosynthesis” was predicted to be upregulated in *Mlx*KO MEFs and downregulated in the other two lines, whereas the “Tumor Microenvironment Pathway” was predicted to be downregulated in *Mlx*KO MEFs and upregulated in the other two KO lines.

To examine the transcript content and behavior of the above IPA pathways in greater detail, we next performed both unbiased and directed gene set enrichment analyses (GSEA) by mining the Enrichr collection, which contains numerous large transcriptomic databases, including those from the KEGG, MSigDB C2, and Mitoproteome repositories [3,22]. Based upon their normalized enrichment scores (NES) and q-values, the most prominent and representative of the gene sets identified could be grouped into seven broad functional categories (Figure 5D and Appendix A) [21]. Two of these (“Mitochondrial structure/function” and “Ribosome biogenesis/translation”) were previously shown to contain numerous gene sets that were downregulated in murine hepatocytes bearing individual or combinatorial knockouts of *Myc*, *Chrebp*, and *Mlx* KO and upregulated in Myc-driven liver cancers [3,8,11] (Figure 5D, Appendix A). While some of the individual gene set contents comprising these two broad categories differed among the three KO MEF lines, their particularly robust enrichment in *Myc*KO and DKO cells was in general agreement with their impaired energy metabolism and translation, whereas their less pronounced downregulation in *Mlx*KO MEFs was consistent with the less prominent impact on these phenotypes (Figure 2D–G and Figure 3F) [3,11]. The five additional broad categories of gene sets that potentially explained other KO MEF behaviors included those associated with cell-cycle regulation, aging, senescence, DNA damage response/DNA repair, and cholesterol metabolism (Figure 5D and Appendix A). PathView analysis of individual gene sets from these categories, selected from among those available in the KEGG database [40], confirmed the distinct identities and overall direction of change of the individual transcripts within each KO line (Appendix A). One particularly illustrative heatmap of a gene set from the cell-cycle regulation category (“KEGG Cell Cycle”; Appendix A) showed the pronounced downregulation of approximately one-third of its 121 members in growth-arrested *Myc*KO and DKO MEFs. This subset was enriched for transcripts encoding minichromosome maintenance (Mcm) proteins (Mcms 2–7), cell division cycle (Cdc) proteins (Cdc 6, 7, 14a, 16, 25a–c, 26, 27), cyclins (A2, B1, B2, E1, E2), cyclin-dependent kinases (Cdk1, 2, 4, 7), and origin recognition complex subunits (Orc1, 2, 3, 4, 6) (Figure 5E).

Finally, to determine how many members of the enriched transcripts shown in Figure 5D and Appendix A originated from genes that are direct targets for Myc, Mlx, or both factors, we again used ChIP data from the ENCODE database [23]. Of the 3714 unique enriched transcripts represented in the above seven categories (Figure 5D), 67.1% (range 58.2–88.5%) and 30.2% (range 25.9–44.0%) were identified as being direct Myc and Mlx targets, respectively, while 26.2% (range 20.7–20.9%) were identified as being dual Myc and Mlx targets (Appendix A). These results confirm our previous findings that approximately one-third of differentially expressed transcripts in *Myc*KO and/or *Mlx*KO hepatocytes are encoded by genes that bind Myc and/or Max in cell-type- and/or context-dependent ways [3].

### 3.6. All KO MEFs Have Multiple but Distinct Defects in DNA Damage Recognition and Repair Pathways

Acquired defects in the response to and repair of DNA damage accompany normal senescence and aging, while telomere dysfunction and the activation of associated senescence pathways occur in some cases of idiopathic pulmonary fibrosis, cirrhosis, and aplastic anemia [41,42,43]. Inherited defects in these pathways also underlie numerous premature aging disorders, known collectively as progeroid syndromes and telomeropathies [33,43]. Consistent with this, chronic exposure to X-rays, the induction of double-stranded DNA breaks (DSBs) by the ectopic expression of restriction endonucleases, and the treatment of pediatric cancer patients with genotoxic chemotherapies can accelerate senescence and aging [44,45,46]. The regulation of some DNA and chromosomal damage recognition/repair pathways has been tied to the extended Myc network and could explain the numerous dysregulated gene sets involving these functions in all three KO MEF lines (Figure 5D and Appendix A) [11,18,47,48].

The p53 tumor-suppressor protein is normally expressed at low levels, with its partial and Myc-dependent cytoplasmic sequestration further restraining its transcriptional activity in the absence of DNA damage [49,50]. Indeed, p53 was barely detectable in WT MEFs but was 1.5–2.5 times more abundant and more nuclearly localized in KO MEFs (Figure 6A,B). Also elevated to variable degrees in KO MEFs were the number and/or intensity of foci for several other members of DNA damage recognition/repair pathways such as 53BP1, γH2AX, Rad51, and Ku80. Finally, biotinylated nucleotide incorporation, catalyzed by terminal deoxynucleotidyl transferase (TdT) and—like γH2AX—an indicator of DSBs (TUNEL assay) [51], was more prominent in KO cells. Exposure of WT MEFs to the DSB inducer etoposide increased TP53 intensity and its nuclear localization in a dose-dependent manner, while similarly increasing the number, size, and/or intensity of foci for other DNA damage response/repair proteins (Figure 6B and Appendix A). In contrast to these well-coordinated dose–response behaviors, those in KO MEFs were more erratic and uncoordinated, reflecting the dysregulated expression patterns of basal DNA damage recognition/repair-related transcripts (Figure 5C–E, Appendix A). Collectively, these findings indicated that each of the KO MEF lines was subject to greater baseline genotoxic stress than WT MEFs and responded to further damage in an abnormal manner.

### 3.7. Immortalized MEFs Proliferate despite the Absence of Myc and Show Differential Chromosomal Instability and Resistance to DNA Damage

As shown above and elsewhere, the genetic or pharmacological inhibition of Myc and/or Mlx impacts a number of cellular functions in distinct ways that also correlate with altered gene expression profiles [2,10,11,52]. Determining each network’s contributions to these changed activities, however, can be challenging given that they are influenced by cell type, the expression level of each factor, whether the studies are performed in vitro or in vivo, and whether the cells are transformed [2,28,53,54]. In the primary MEFs used here, this was further compounded by the inevitable onset of senescence and proliferative arrest that accompanies in vitro passage, as well as by the more immediate cell cycle arrest than follows in the aftermath of *Myc* excision (Figure 1E,G) [16,33,43]. Thus, to better appreciate the contribution(s) of each network in the absence of these dynamic and confounding variables, we generated SV40 T-antigen-immortalized cell lines from each parental floxed MEF strain and then excised the target genes [55].

Immortalized MEFs differed from their primary counterparts in several important respects. For example, despite target gene excision efficiencies rivaling those of primary cells (Appendix A), all three immortalized KO cell lines continued to proliferate, albeit at rates somewhat slower than those of WT MEFs (Figure 7A). SV40 T-antigen-mediated immortalization, which is associated with the inactivation of both p53 and the retinoblastoma (Rb) tumor suppressors [55], was therefore functionally equivalent to *Mlx* KO in primary MEFs, in that it rescued the growth-arrested state of *Myc*KO cells.

Immortalized *Myc*KO cells, grown to confluence and then maintained continuously for 2 weeks without subsequent sub-culturing, achieved saturation densities only around half those of similarly maintained WT cells, whereas *Mlx*KO cells achieved densities more than twice those of WT cells, and DKO cells achieved intermediate densities (Figure 7B). The higher saturation density of *Mlx*KO cells was also readily apparent upon crystal violet staining of monolayer cultures and by phase-contrast microscopy (Figure 7C,D). The latter showed that, while WT, *Myc*KO, and DKO cultures were notable for the cobblestone-like appearance that accompanies the growth arrest of other contact-inhibited fibroblasts, *Mlx*KO cultures clearly lacked this highly contact-inhibited state.

Repeat cell-cycle analysis unexpectedly showed > 80% of immortalized *Myc*KO cells to have acquired a tetraploid or pseudo-tetraploid state within 2 weeks of their derivation (Figure 7E–G). The inactivation of p53 and/or Rb by SV40 T antigen in *Myc*KO cells was therefore both necessary and sufficient to drive the rapid emergence and maintenance of this additional chromosome-level genomic instability in the face of pre-existing DNA damage response dysfunction (Figure 5C–E and Figure 6). The maintenance of stable euploidy in DKO cells indicated that the transition to tetraploidy by *Myc*KO cells was abetted by an intact Mlx network.

The ability to maintain continuously proliferating immortalized MEF cell lines afforded the opportunity to determine whether their DNA damage recognition/repair defects were associated with differential susceptibilities to additional genotoxic insults (Figure 6, Appendix A). While regulating many genes involved in DNA damage recognition and repair, p53 and Rb also integrate these with senescence, aging, and apoptosis pathways [42,43,49,50,56,57]. We thus hypothesized that the immortalization of KO MEFs, coupled with pre-existing extended-Myc-network-related DNA repair defects—particularly in *Myc*KO and DKO MEFs (Figure 5D, Appendix A)—would render them less susceptible to senescence and/or apoptosis and, thus, more tolerant of additional DNA damage. To test this, we examined several chemotherapeutic and physical agents with distinct and, in some cases, overlapping mechanisms of action. These included the induction of DSBs (etoposide and mitomycin C), single-stranded breaks (6-thioguanine), alkylation (busulfan and mitomycin C), methylation (temozolomide) and the formation of inter- and intrastrand crosslinks (cisplatin and UV light, respectively) (Figure 7H–N). As predicted, *Myc*KO and DKO MEFs—and to a lesser extent *Mlx*KO MEFs—were differentially resistant to these agents. Collectively, these results indicated that the distinct gene expression patterns associated with DNA damage recognition and repair were correlated with actual survival differences in response to a spectrum of genotoxic insults.

Because the above studies were performed with immortalized MEFs, we anticipated that their gene expression profiles—particularly those pertaining to DNA damage recognition and repair pathways—would differ substantially from, yet overlap with, those of their non-immortalized counterparts (Figure 5). This would likely reflect the sustained proliferation of all four immortalized MEF cell lines, the newly acquired *Myc*KO MEF tetraploidy, and the consequences of p53 and Rb inactivation, which by themselves can alter the expression of multiple DNA damage recognition- and repair-associated genes [50,56,58] (Figure 7E–G). We therefore repeated RNA-Seq on each of the immortalized cell lines. In addition to the originally described DNA damage recognition and repair pathway-related transcripts in primary MEFs (Figure 5D, Figure 8A,B and Appendix A), newly acquired signatures of dysregulation could now be seen.

Several additional major functional categories of gene sets were enriched in immortalized KO cells (Figure 8C–E and Appendix A). As was true for primary cells (Figure 5D), these overlapped among all three KO groups, and the expression levels of individual transcripts were distinct even within shared gene sets. Absent from these top functional categories in the immortalized cells were those associated with aging, senescence, and the cell cycle, likely reflecting the cells’ immortalized and proliferative state. Also of less prominence were gene sets related to mitochondrial and ribosomal structure and function. This may again be related to the more highly proliferative nature of immortalized *Myc*KO and DKO cells, which require increases in both energy generation and protein translation to meet their higher proliferative demands.

A rebalancing in the expression of other extended Myc network members—particularly those with previously suspected tumor-suppressor-like functions—might explain the Myc-independent growth of immortalized *Myc*KO and DKO MEFs (Figure 4A), as well as more general contributions to the immortalized state [2]. The dysregulation of several Myc target genes can restore the proliferative and/or mitochondrial defects of *Myc-/-* cells to variable degrees [37,38,59,60]. We examined 18 of the above transcripts among all eight primary and immortalized cell lines and found distinct expression patterns of expression for each. Among the most noteworthy of these were the downregulation of Max and Mxd2 transcripts in immortalized *Myc*KO and DKO cell lines, the upregulation of Hmga1 and downregulation of Mxd4 in immortalized *Myc*KO MEFs, and the downregulation of Plac8/Onzin across all KO cell lines—particularly immortalized ones (Figure 8F). These results show that each KO cell line uniquely readjusts its expression of other extended Myc network members, that many of these signatures in otherwise isogenic lines are altered by immortalization, and that no single pattern of expression readily explains the ability of *Myc*KO and DKO MEFs to proliferate in a Myc-independent manner.

We also asked whether the eventual Myc-independent resumption of growth of primary DKO MEFs might be related to the inhibition of the p53 and/or Rb pathways that contributed to the proliferation of SV40 T-antigen-immortalized *Myc*KO cells (Figure 7A). Accordingly, we reanalyzed our original RNA-Seq data from growth-arrested primary *Myc*KO and DKO cells (Figure 1E,G). Among their ~2300 gene expression differences (*q* < 0.05), the top categories identified by GSEA consisted of transcripts involving the p53 and Rb pathways and the G2/M checkpoint. Moreover, DKO MEFs, despite having been growth-arrested at the time of the analysis, showed enrichment of these transcripts in directions that favored a proliferative phenotype (Appendix A). We concluded that the eventual resumption of proliferation by primary DKO cells may also involve the same general pathways targeted by the SV40 T antigen during immortalization. Whereas the latter inhibits TP53 and Rb directly, Mlx loss involves the inhibition of some of these tumor suppressors’ downstream targets. The presumed relaxation of the G2/M checkpoint seen in DKO MEFs may also be involved in preparing for cell-cycle re-entry between days 14 and 21 (Figure 2A). Despite the ability of Mlx loss to alter the expression of genes in the p53 and Rb pathways and overcome cell-cycle arrest, this only partially substitutes for the SV40 T antigen, as the cells—along with the other three MEF cell lines—do eventually enter a state of permanent replicative arrest.

## 4. Discussion

Myc’s varied roles in proliferation and senescence are well documented and mechanistically linked. In the former case, they depend on the degree to which it is over- or underexpressed, whereas in the latter case they also depend on the identity and duration of the pro-senescent stimulus, the presence of coexisting mutations, the cell type being studied, and whether it is also transformed [16,28,54,60]. The incremental depletion of Myc in early-passage primary MEFs causes an increasingly pronounced G_0_/G_1_ cell-cycle arrest that is—at least initially—unassociated with alterations in cell size or senescence [16]. Tumor cells also rapidly cease proliferating in response to acute Myc depletion, although whether cell-cycle arrest occurs in G_0_/G_1_, S, or G_2_/M depends on the presence of coexisting cell-cycle defects [28]. Immortalized *Myc^−/−^* rat fibroblasts, which proliferate very slowly, increase their size and markedly shorten their doubling time following Myc re-expression [10,15,38]. Oncogene-induced senescence—notably, that promoted by mutant forms of Ras and B-Raf—can be suppressed by overexpressing Myc; this is crucial to circumventing a major block to tumor progression while concurrently driving aberrant cell-cycle progression and tumor-associated genomic instability [2,18,47,54].

In contrast to a previous study of primary *Myc^−/−^* MEFs [16], the *Myc*KO and DKO cells described in the present work were significantly larger than WT and *Mlx*KO cells and contained a greater proportion of cells with high side-scatter attributable to the increased lysosomal content that often accompanies senescence (Figure 3A) [33,43]. These cell lines also displayed increasingly prominent G_0_/G_1_ arrest over the course of at least one week (Figure 2A,B). Other signs of senescence and aging—such as increased uptake of 2-NBDG, more prominent SA-β-gal activity, mitochondrial dysfunction in the form of ROS generation, and neutral lipid accumulation—also emerged during this time, although not necessarily in parallel (Figure 2D–F and Figure 3B,C). These findings reinforce the idea that the properties and behaviors of senescent cells may be both transient and cell-type-specific, with no single one sufficing to define the state [31,33,35,53,61]. Differences in the behaviors of *Myc*KO MEF properties reported here and elsewhere [16] may reflect when target gene excision occurred (i.e., at the time of conception in vivo versus postnatally in vitro) and/or the longer time-course of our studies.

The mitochondrial dysfunction seen in *Myc*KO and DKO MEFs was not unexpected, given that related defects have been described in other normal and cancer cell types following Myc and/or Mlx loss or inhibition [3,10,14,25]. For example, the exceedingly slow proliferation of *Myc^−/−^* rat fibroblasts has been attributed to atrophic mitochondria and ETC compromise that produce only one-third the normal level of ATP [10]. These defects can be rescued and normal growth rates restored following Myc re-expression, although this requires several days [10]. Primary hepatocytes lacking *Myc* and/or *Mlx*, along with experimental hepatoblastomas originating from them, show variable degrees of aberrant mitochondrial structure and function, the suppression of genes needed for their maintenance, the accumulation of neutral lipids, and proliferation [3,14]. The seemingly normal TCA cycle substrate utilization of our *Myc*KO and DKO MEFs, on the other hand, was sustained only at the expense of a near-doubling of mitochondrial mass (Figure 2C,G). Despite this compensatory action, mitochondrial dysfunction persisted, as evidenced by the generation of ROS and the accumulation of neutral lipids (Figure 2E,F). The latter has been attributed to a switch from glucose to fatty acid utilization as the preferred energy source, with the uptake of the latter exceeding the actual energy requirements and the excess being stored as neutral lipids [3,14,25].

Initial transcriptomic analyses provided molecular correlates to the mitochondrial, translational, and cell-cycle defects in the MEFs reported here and for other previously described cell types (Figure 5) [3,8,25]. They confirmed that, while the Myc and Mlx networks each control unique target genes, as many as 20–30% are co-regulated in ways that reflect tissue type, the prevailing amounts of Myc and Mlx, the degree to which these factors compete for common binding sites, and the cell’s metabolic landscape [2,3]. This is consistent with our finding that, whereas *Myc*KO, *Mlx*KO, and DKO MEFs dysregulate general categories of gene sets that can be readily linked to specific cellular functions and phenotypes, the identities of the enriched gene sets comprising these categories are distinct (Figure 5, Figure 8 and Appendix A). That the expression of individual transcripts within the common GSEA sets may also differ provides additional context within which to interpret other behavioral subtleties.

Compelling evidence in support of this notion derives from the multiple gene sets and selected proteins in the “DNA damage recognition/repair” category that were dysregulated by primary KO MEFs (Figure 5D, Figure 6, Appendix A). However, observing the consequences of this in response to genotoxic agents required that all starting cell populations remain proliferation-competent and free of the continuously changing constraints imposed by aging and senescence (Figure 7). SV40 T-antigen-mediated immortalization provided a reasonable means by which the sensitivity to DNA damage among the MEF groups could be assessed on such an equal footing, although it did introduce additional gene expression differences associated with p53 and Rb inactivation (Figure 8) [50,56]. Cell-line-specific responses were anticipated based on differences in ploidy, gene expression profiles, and the behaviors of several DNA damage recognition proteins (Figure 5D, Figure 6, Figure 7H and Figure 8A–E). Indeed, the *Myc*KO and DKO lines in particular tended to demonstrate resistance to mechanistically diverse DNA-damaging agents (Figure 7). These results contrast with those seen in cell lines derived from patients with defective DNA damage recognition/repair pathways, such as Fanconi anemia and Nijmegen breakage syndrome, which are exceedingly sensitive to DNA damage [62,63]. These discrepancies likely have several non-mutually exclusive explanations. First, patient-derived cell lines bear defects in single genes and pathways, whereas our KO cell defects were widespread and involved multiple and often interconnected pathways. Second, p53 and Rb—both of which were inactivated in our cell lines but not in patient-derived cell lines—also participate in normal DNA repair and communicate with the extended Myc network [57,64]. Indeed, by analyzing murine fibroblast ChIP-Seq data (TP53 23651856 ChIP-Seq MEFs Mouse https://maayanlab.cloud/Enrichr/enrich?dataset=433056a9d8a9fec5170d64d872b0788b (accessed on 14 August 2022)) in the Enrichr database [22], we found that ~18% of the 5589 genes whose transcripts are dysregulated in immortalized *Myc*KO MEFs bind p53, and that ~39% of the 2595 genes that bind p53 are dysregulated in response to *Myc*KO (not shown). Third, tetraploidy—such as that seen in immortalized *Myc*KO cells—is commonly associated with its own DNA repair defects [65]. Finally, the large number of genes that are dysregulated in response to *Myc* and/or *Mlx* inactivation—particularly those pertaining to aging and senescence—may impact survival in ways that are independent of classic DNA damage recognition and repair pathways. Collectively, our results support the notion that *Myc*KO and/or *Mlx*KO cells are more tolerant of genotoxic insults by virtue of possessing multiple defective DNA damage recognition and repair pathways that are reinforced by concurrent inactivation of p53 and Rb. The ensuing loss of coordination among these pathways leads to largely ineffectual apoptotic responses.

Another question is why, following their initial proliferative arrest, primary DKO MEFs re-enter log-phase growth (Figure 1 and Figure 4A). This is clearly related to the concurrent loss of *Mlx,* which is the only engineered feature that distinguishes these cells from permanently arrested *Myc*KO cells. In mice, *Mlx* is a suppressor of benign hepatic adenomatosis, and immortalized *Mlx*KO cells achieve higher saturation densities than WT, *Myc*KO, or DKO MEFs (Figure 7B–D) [3]. Further supporting a tumor-suppressor-like role for Mlx are its partial silencing in immortalized *Myc^−/−^* rat fibroblasts, its direct correlation with Myc levels in some cancers (Figure 4J,K) [10,15], and its regulation of genes involved p53- and Rb-mediated growth suppression and G_2_/M blockage (Appendix A). Together with the transcriptional rebalancing of other extended Myc network members and downstream Myc targets, our findings indicate that certain normal cellular functions—most notably proliferation—can be maintained in Myc’s absence, although the precise changes needed for this appear to be highly context-dependent.

An equally important question is why primary DKO MEFs eventually lose their senescence-like state and become tetraploid following immortalization (Figure 4 and Figure 7H). While senescence is normally perceived as a state of permanent growth arrest, it is sometimes reversible when p53 and Rb are compromised, and it may be an important means by which cancer cells circumvent oncogene-induced senescence [66]. Myc overexpression and p53 inactivation cooperatively promote tetraploidy in a manner that is amplified by mitotic spindle dysfunction [18]. The restoration of p53 in Myc-overexpressing euploid cells prior to their development of tetraploidy promotes robust apoptosis, indicating that irreparable genomic and/or mitotic spindle damage is an early event that precedes any changes in chromosomal content [18]. The rapid induction of tetraploidy in both Myc-overexpressing and immortalized *Myc*KO cells suggests that this chromosomal instability is driven less by the actual level of Myc than by the pre-existing DNA damage and cell-cycle dysregulation (Figure 2A, Figure 5D,E, Appendix A).

Both in vitro and in vivo, cells and tissues undergoing normal and pathological aging and senescence display many of the features and gene expression changes that we have identified in KO MEFs—particularly those lacking Myc. These include the gradual loss of mitochondrial and translational efficiency, the accumulation of fat, and a reduced ability to respond to and/or repair DNA damage [26,42,67,68]. Age is the single strongest predictor of cancer in numerous species, and strong links exist between premature aging, senescence, DNA repair defects, and early-onset cancer [33,43,62]. For example, individuals who received genotoxic forms of chemotherapy for cancer as children often manifest signs of premature aging [44].

In mice, *Myc* haploinsufficiency is associated with an extended and healthier lifespan and, although the precise reasons for this have not been clearly defined, it may be related to a reduced cancer incidence [52]. Because *Myc*KO homozygosity is embryonically lethal, it has not been possible to determine the degree to which its consequences differ from those associated with heterozygosity [13]. The relatively few gene expression differences between *Myc^+/+^* and *Myc^+/−^* tissues likely reflect Myc’s low basal expression in the former, as well as the fact that target genes with the highest-affinity Myc-binding sites, are not significantly impacted by 50% reductions in Myc levels [52]. Many of the gene sets enriched in the KO tissues described here have been identified in the livers of *Myc*KO, *MlxKO*, and DKO mice, which also demonstrate variable defects in proliferation, mitochondrial and ETC function, and neutral lipid metabolism [11,14]. Collectively, our findings raise the question of how the body-wide elimination of these genes postnatally might impact long-term survival and associated phenotypes.

To fully appreciate the relationships among the various defects and phenotypes of the MEFs described in the present work, it is important to understand that virtually all of the functional categories impacted by *Myc* and/or *Mlx* KO (Figure 5D and Appendix A) also crosstalk with one another in ways that are Myc- and/or Mlx-network-independent as well (Figure 8G). For example, although *Myc* and *Mlx* KO directly impact mitochondrial and ribosomal structure and function (Figure 2 and Figure 5D) [2,3,8,9,11,14], so too do normal aging and senescence [32,33,67,68,69,70] The increased ROS and decreased ATP produced during such times can compromise functional categories such as those involving genomic integrity and translation, which are then further impacted more directly at the level of individual genes by loss of *Myc* and/or *Mlx*. Thus, the ultimate behaviors and properties of KO MEFs reflect the direct consequences of *Myc* and *Mlx* gene inactivation, as well as the loss of normal inter-network communication, balance, and cooperation.

## Figures and Tables

**Figure 1 cells-11-04087-f001:**
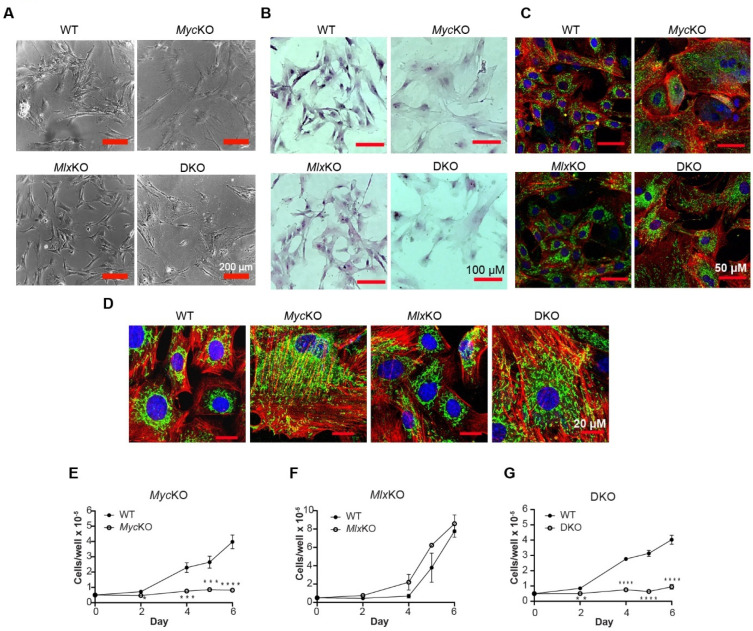
Morphological features and growth curves of MEFs: (**A**) Phase-contrast images of live, day 14 MEFs. (**B**) The same cells grown on coverslips, fixed, and stained with H & E. (**C**) Confocal micrographs of day 14 MEFs after staining of nuclei with Hoechst 33342, mitochondria with MitoTracker™ Green, and the cytoskeleton with Cell Mask Orange Actin Tracker. (**D**) Higher-magnification live confocal images of the indicated cells obtained on day 21, showing an apparent increase in mitochondrial mass in *Myc*KO and DKO MEFs (green). In A-D, red bars are included as sizing references as indicated in the rightmost/lower rightmost images (**E**–**G**) Growth curves of *Myc*KO (**E**), *Mlx*KO (**F**), and DKO MEFs (**G**), each performed in parallel with WT cells. All cells were plated in 4OHT-free medium on day 7. Each point represents the mean of three replicas ± 1 S.E. Significance was determined by multiple ratio paired *t*-tests; * *p* < 0.1, ** *p* < 0.01, *** *p* < 0.001, **** *p* < 0.0001.

**Figure 2 cells-11-04087-f002:**
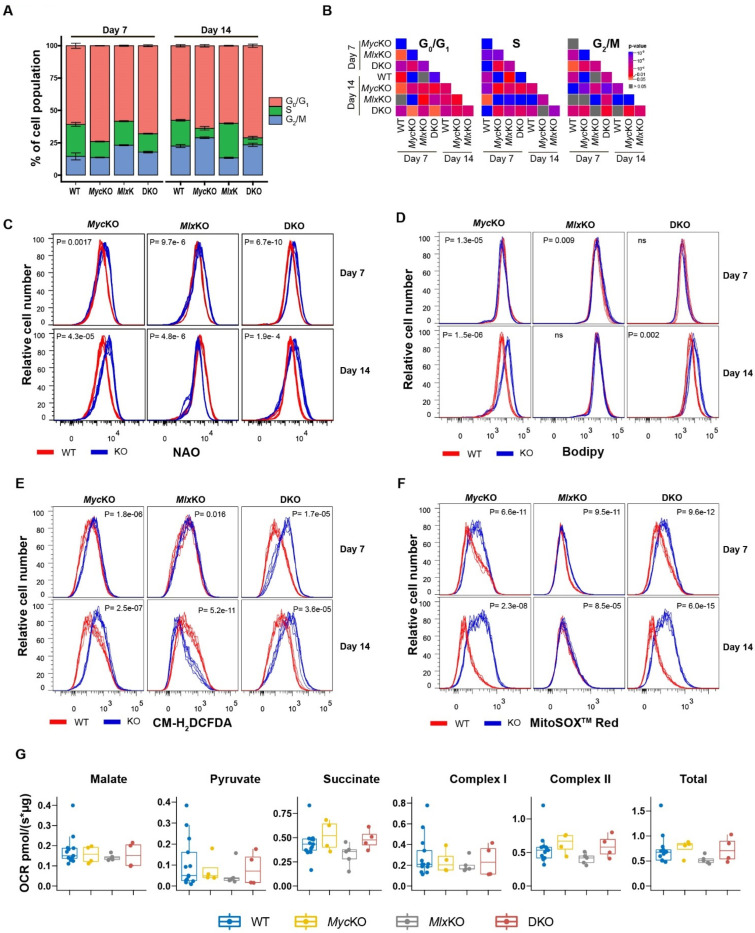
Cell cycle and mitochondrial functional profiling in primary MEFs: (**A**) Cell-cycle analysis. Six replicates of the indicated day 7 and day 14 MEFs were re-plated in the absence of 4OHT, harvested two days later at ~60% confluence, stained with propidium iodide, and assessed by flow cytometry. The results of each experiment are depicted as histograms, which show the mean fraction of cells in G_o_/G_1_, S, and G_2_/M ± 1 S.E. (**B**) The percentages of cells in G_0_/G_1_, S, and G_2_/M on day 7 and day 14 shown in (**A**) were compared in order to identify significant inter- and intragroup differences. Gray boxes = not significant. (**C**) Mitochondrial mass of the indicated cell lines on days 7 and 14, as determined by NAO staining (*n* = 6). (**D**) Neutral lipid contents of each cell line on days 7 and 14 were determined by BODIPY^TM^ 505/515 staining (*n* = 6). (**E**) Quantification of total ROS production on days 7 and 14 obtained by staining with CM-H_2_DCFDA (*n* = 6). (**F**) Quantification of mitochondria-specific O_2_^−^ production on days 7 and 14 obtained by MitoSOX™ staining (*n* = 6). In (**E**,**F**), results for each of the samples are shown. (**G**) Oxygen consumption rates (OCRs) of WT and KO MEFs in response to the indicated TCA cycle substrates and activities of complex I, complex II, and complexes I + II (total). Day 10 cells were permeabilized with digitonin and exposed to the indicated substrates while quantifying OCRs as previously described [19].

**Figure 3 cells-11-04087-f003:**
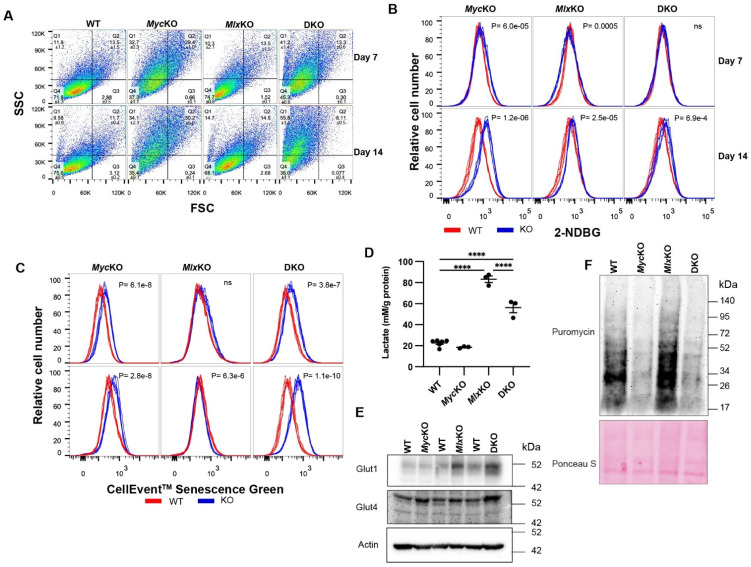
Variable senescence-like features of KO cells: (**A**) Side (SSC) and forward (FSC) light scatter of the indicated cell lines on days 7 and 14. Six replicates were performed for each cell line, with the mean percentages and SEs indicated in each quadrant. (**B**) Uptake of the glucose analog 2-NBDG. (**C**) Cleavage of the SA-β-Gal substrate CellEvent™ Senescence Green. In (**B**,**C**), the curves for six replicas are shown, with *p*-values between each group shown in the upper-right corner of each panel. (**D**) Lactate levels in quiescent MEFs. The indicated cell lines were re-plated on day 10 at densities high enough to allow replicative arrest of WT and *Mlx*KO cells to be achieved over the next 1–2 days. Non-replicating *Myc*KO and DKO cells were plated at densities that allowed them to achieve confluence by the next day. All cultures were then maintained for three additional days in medium containing 0.2% FCS to further consolidate quiescence, at which point lactate levels were quantified and normalized to total protein [19]. Significant differences are expressed as for Figure 1E–G. **** *p* < 0001. (**E**) Expression of the glucose transporters Glut1 and Glut4 in MEFs measured by immunoblotting. (**F**) Rates of protein synthesis in the indicated MEFs. Subconfluent cultures of d10 MEFs were labeled with puromycin for one hour, lysed, and subjected to SDS-PAGE before probing with an anti-puromycin antibody. Equal protein loading was confirmed by Ponceau red staining.

**Figure 4 cells-11-04087-f004:**
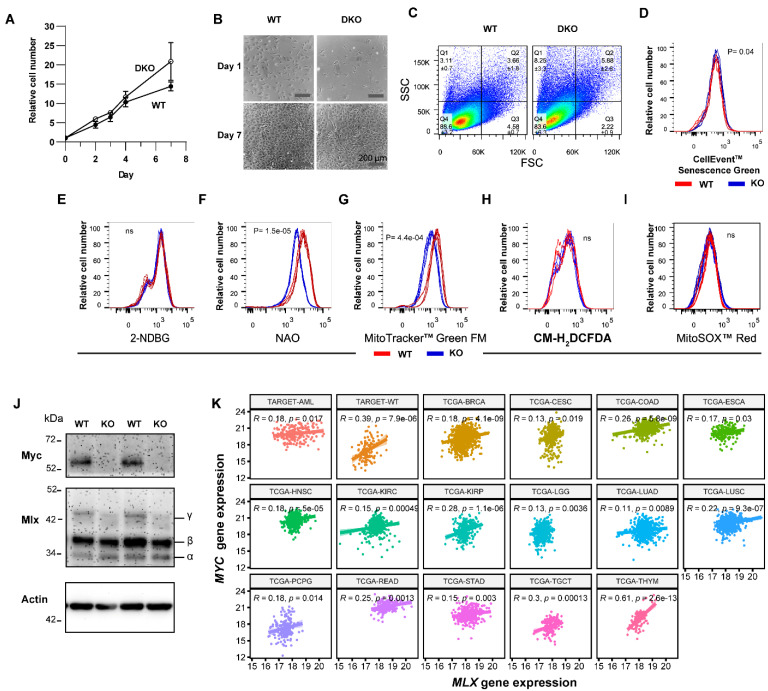
Correlations between Myc and Mlx expression: (**A**) WT and DKO MEFs were maintained until day 28 and then re-plated at low density. Growth rates and *p*-values were determined from three replicates at the indicated times, as described in Figure 1E–G. (**B**) Phase-contrast micrographs of WT and DKO MEFs on days 29 and 36 (i.e., 1 and 8 days after re-plating, respectively). (**C**) Side- and forward-scatter of WT and DKO cells performed on ~day 43, as described in Figure 3A. (**D**) CellEvent^TM^ Senescence Green staining performed on ~day 43, as described in Figure 3C. (**E**) 2-NBDG uptake by ~day 43 WT and DKO MEFs, performed as described in Figure 3B. (**F**,**G**) NAO and MitoTracker™ Green staining, respectively, performed on ~day 43–45 MEFs, as described in Figure 2C. (**H**,**I**) CM-H2DCFDA and MitoSOX staining, respectively, performed on ~day 43–45 MEFs, as described in Figure 2E,F. (**J**) Selective loss of the γ isoform of the Mlx protein in immortalized *Myc^−/−^* rat fibroblasts, and its retention in the parental *Myc^+/+^* cells from which they were derived [15]. (**K**) Correlations between Myc and Mlx transcript levels in 17 different cancer types from TCGA and the pediatric cancer TARGET collections. Pearson’s correlation coefficients and *p*-values are shown within each panel.

**Figure 5 cells-11-04087-f005:**
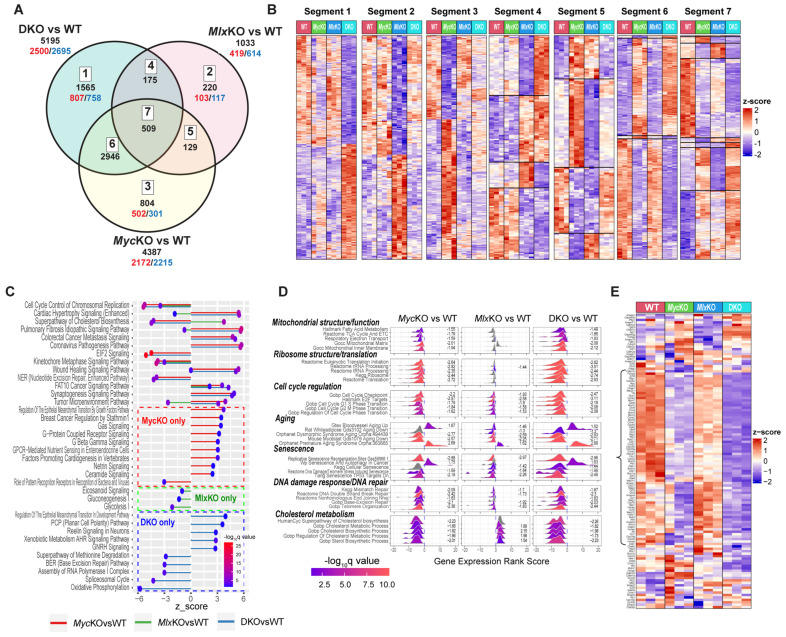
Transcriptional profiling of primary MEFs: (**A**) Differentially expressed transcripts in each KO group of MEFs relative to WT MEFs. Numbers within white boxes (1–7) identify specific segments of the Venn diagram referenced in the text. Numbers in black beneath these boxes indicate the total number of dysregulated transcripts in that segment relative to the WT, with those in red and blue indicating up- and downregulated transcripts, respectively. (**B**) Heatmaps of transcript groups from each of the segments shown in **panel A**. The results of three biological replicates of RNA-Seq experiments are shown for each of the indicated MEF groups. (**C**) IPA profiling of broad functional pathways represented by transcripts from different segments of the Venn diagram shown in (**A**). Only pathways with significant z-scores are shown. (**D**) GSEA performed with 6347 transcripts from (**A**) using the Enrichr database collection [22]. ClusterProfiler was used to display representative examples of the gene sets within each general category using the ridgeline plot application tool [21]. See Appendix A for additional GSEAs from each of these broad categories. (**E**) Heatmap of individual transcripts from the 121-member KEGG_CELL_CYCLE gene set taken from the “Cell Cycle Regulation” category in (**D**). The bracket indicates a transcript subset that was particularly downregulated in non-replicating *Myc*KO and DKO MEFs and less so in *Mlx*KO MEFs.

**Figure 6 cells-11-04087-f006:**
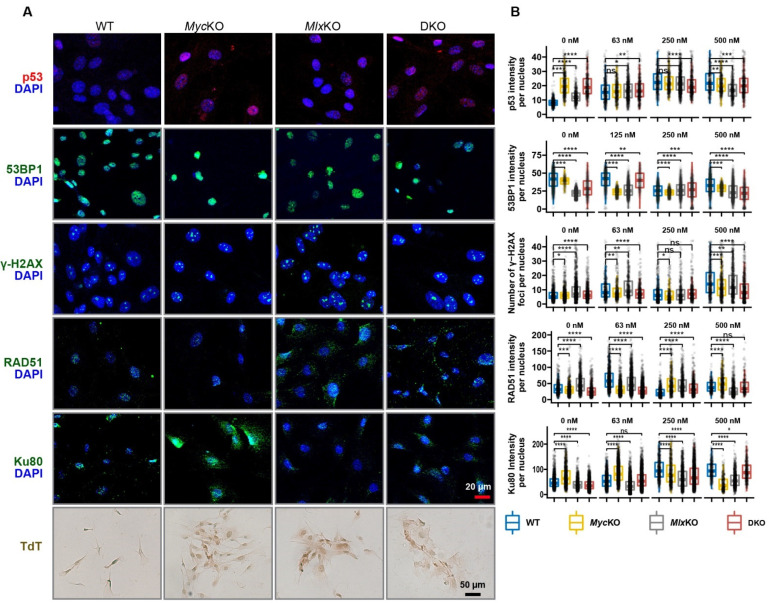
DNA damage response/repair defects in KO MEFs: (**A**) Baseline expression of DNA damage recognition/repair proteins detected by immunohistochemical staining in the indicated primary MEFs. A TdT/TUNEL assay was used as an alternate way to identify double-stranded breaks (DSBs). (**B**) Response of the proteins shown in (**A**) to DSBs induced by etoposide. MEFs were plated as described in (**A**), allowed to attach for 24 h, and then exposed to the indicated concentrations of etoposide for 16 h, fixed, and stained for the indicated proteins. Staining intensities and/or numbers of foci were then quantified for each drug concentration and compared to untreated cells. For each concentration of etoposide, images from at least ~700–800 individual cells were quantified using ImageJ [14]. See Appendix A for the results shown in (**panel B**) depicted as dose–response curves. * *p* < 0.1, ** *p* < 0.01, *** *p* < 0.001, **** *p* < 0.0001.

**Figure 7 cells-11-04087-f007:**
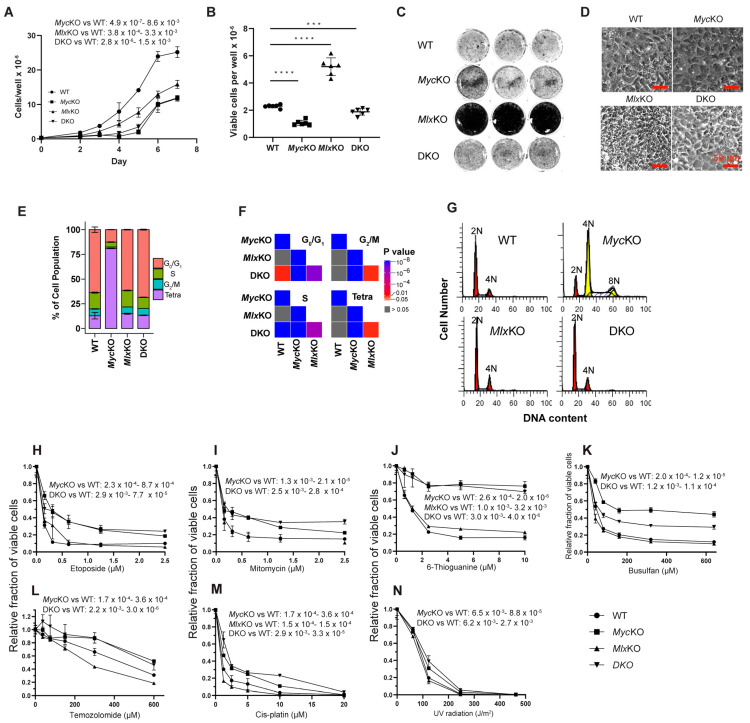
Properties of SV40 T-antigen-immortalized MEFs: (**A**) Growth curves performed as described in Figure 1E–G. (**B**) Maximum saturation densities attained by the indicated immortalized MEF lines. Logarithmically growing cells were allowed to achieve a confluent state over 2–3 days and were then maintained for an additional 2 weeks, with standard medium changes (10% FBS) being performed twice weekly. Cell counts were performed on six replicates from each group. (**C**) The indicated cells were plated and maintained under confluent conditions for two weeks as described for (**B**) and then stained with crystal violet. Three replica wells are shown for each group (**D**). Phase-contrast micrographs of typical fields obtained just prior to crystal violet staining of the cells shown in (**C**). (**E**) Cell-cycle analyses performed and analyzed as described in Figure 2A,B. The results shown are the mean values obtained from six independent analyses performed in parallel for each cell line ± 1 S.E. (**F**) Comparison of cell-cycle populations among the four MEF groups, as described in Figure 2B. (**G**) Typical cell-cycle profiles of each MEF cell line showing the presence of tetraploidy/pseudo-tetraploidy in > 80% of *Myc*KO cells. (**H**) Dose–response curves of the indicated cell lines maintained in the indicated concentrations of etoposide. Cells were plated at densities of 4–6 × 10^4^ cells/well, allowed to attach for 24 h, and then exposed to fresh medium containing the indicated concentrations of etoposide. Total viable cell counts were determined three days later. Each point represents the mean of three replicates ± 1 S.E. (**I**–**M**) Dose–response curves performed as described in (**H**) except with the indicated drugs. (**N**) Dose–response curves performed as described in (**H**) except that the cells were exposed to the indicated doses of UV light. *** *p* < 0.001, **** *p* < 0.0001.

**Figure 8 cells-11-04087-f008:**
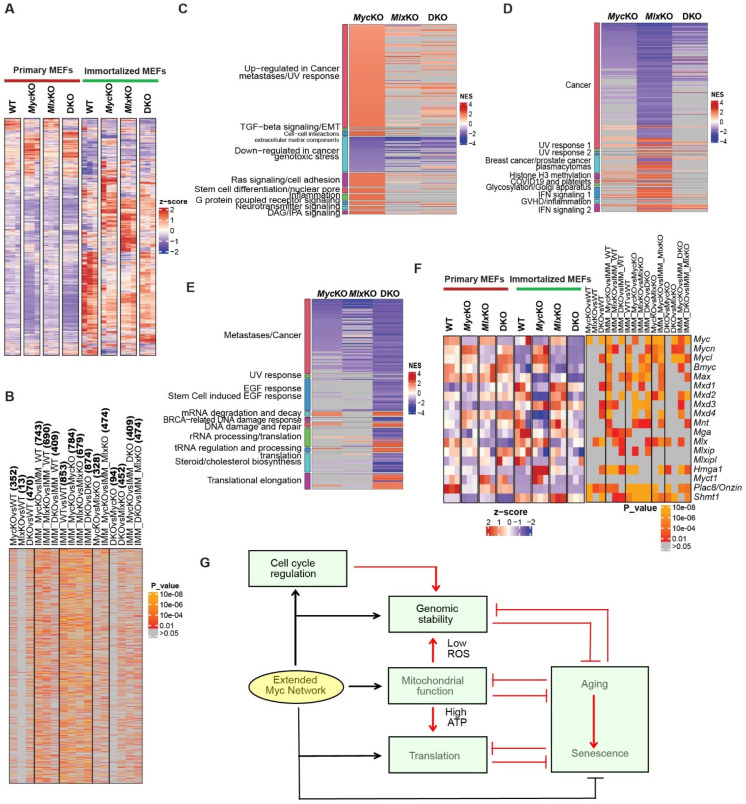
RNA-Seq analyses of immortalized MEFs: (**A**) Expression profiling (z-scores) of all 1519 unique transcripts encoding proteins involved in DNA damage recognition/repair from the m MSigDB C2 database in primary and immortalized MEFs. (**B**) Significant differences in the expression levels of the transcripts depicted in (**A**) among pairwise comparisons of primary and immortalized MEFs. The pairwise comparisons represented by each column are indicated at the top of each column, with the number of significant gene expression differences (*q* < 0.05) shown in bold. Note that all four immortalized MEF cell lines contain significantly more gene expression differences relative to their WT counterparts (columns 7–10). (**C**) Major categories of gene sets enriched in immortalized *Myc*KO MEFs. GSEA was performed using gene sets obtained from the MSigDB C2 database and clustered using the Bioconductor clusterProfiler tool (https://bioconductor.org/packages/release/bioc/html/clusterProfiler.html (accessed on 14 August 2022)). Heatmaps were drawn using the ComplexHeatmap tool, also from the Bioconductor website (https://bioconductor.org/packages/release/bioc/html/ComplexHeatmap.html (accessed on 14 August 2022)). The top categories of gene sets are depicted at the left of the heatmap. See Appendix A for a full list of each category’s members. Each colored line on the heatmap represents one group of gene sets. Gray lines = not significant for the indicated group. (**D**) Major categories of gene sets enriched in immortalized *Mlx*KO MEFs. GSEA and heatmap construction were performed as in (**C**). See Appendix A for a full list of each category’s members. (**E**) Major categories of gene sets enriched in immortalized DKO MEFs. GSEA and heatmap construction were performed as in (**C**,**D**). See Appendix A for a full list of each category’s members. (**F**) Expression profiling (*z*-scores and *p*-values) of transcripts encoding members of the extended Myc network and several genes whose overexpression (*Shmt, Myct1, Hmga1/HmgI(Y)*) or knockdown (*Plac8/Onzin)* can at least partially rescue the loss of Myc expression in other fibroblast lines [37,38,59,60]. (**G**) Direct and indirect relationships between the extended Myc network and the functions and phenotypes described in the current work. Myc and/or Mlx (yellow oval) maintain the balance among individual gene sets in the broad functional categories shown in green boxes and in Figure 5D (black lines). KO of Myc and/or Mlx causes dysregulation of these gene sets. This, in turn, can impact other categories in ways that are independent of the Myc and/or Mlx networks (red lines).

## Data Availability

All data needed to evaluate the conclusions in this paper are present in the paper and/or the Appendix A. Original sequencing data were deposited in the Gene Expression Omnibus (GEO) website (Access: GSE210362; and secure token for reviewer: wtexgaoqnnkxzuj) (https://www.ncbi.nlm.nih.gov/geo/ (accessed on 1 June 2022)).

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
