# Peer review of "Disruption of Multiple Overlapping Functions Following Stepwise Inactivation of the Extended Myc Network"

_cells, 2022, doi:10.3390/cells11244087_

Round 1

Reviewer 1 Report

Prochownik and collaborators generate single Myc-KO and Mlx-KO MEFs as well as dKO-MEFs to dissect the contribution of these components of the "extended Myc network" in transcription, short-term and long-term proliferation, survival, and metabolism. They engineer both primary MEFs and SV40-immortalized MEFs.

The main criticism that I have is that given the complexity of the extended Myc network, the deletion of two of the components represents a limited genetic investigation that is difficult to interpret and doesn't not allow for any real insight into the role of these two components and their genetic dependencies.

Indeed, despite the wealth of data presented and the articulated analysis, the interpretation and conclusion are highly inferential. In the end this work is highly descriptive and not genuinely informative.

Some basic mechanistic questions remain unaddressed, for instance: how is Mlx loss partially rescuing the loss of Myc? Is it a real rescue or just a long term adaptation or a clonal drift of dKO-MEFs? Also, all the transcriptional analyses are difficult to interpret without teasing out what are the transcriptional changes due to alteration in the direct targets of Myc or Mlx and what changes are a consequence of cell cycle arrest, cellular aging, or senescence. 

Minor point:

The abstract is confusing. It makes several references to KO-MEFs, but it is unclear whether this refers to MycKO, MlxKO or both.

How many independent MEFs isolates were used in this work?

Can ectopic expression of the wild-type cDNA rescue the phenotypes of the KO-MEFs?

Reviewer 2 Report

This manuscript entitled “Disruption of Multiple Overlapping Functions Following Step-Wise Inactivation of the Extended Myc Network” is submitted by Dr. Wang, et. Al. the authors used three knockouts of MycKO, MlxKO and “double KO” (DKO) to study effects on primary and immortalized murine embryonic fibroblasts (MEFs). In general, the study is systematically designed, however, some controls for data analysis can be introduced and conclusions can be written more explicitly.

1.       Title: “….Step-Wise inactivation…..”, this statement only appears in the title and not explain in the main text.

2.       There are many lines underlined. Is this done by intention, it doesn’t seem to help / focus with reading.

3.       Results section can be numerically listed or bold for easy reading.

4.       Figure 1E-G. Growth curves of MycKO (E) and DKO MEFs appears equal / very similar. It is difficult to come to a conclusion of the effect of the DKO without subtract the MycKO effect. 

5.       Has the immortalized cells validated and which passages was used in the study?  Figure s15 of qPCR and WB were for the confirmation of KO.

6.       How to understand the relationship between figure 2a and figure 2b? e.g., day 7 WT vs day 14 DKO, what does that data mean?

7.       It appears in primary MEFs, DKO perform similar to MycKO (e.g., figure 1e, figure 3a), but in immortalized MEF, its performance more close to MlxKO (figure 7a and 7g). please discuss these data.

8.       P value misses in figure 3a.

9.       Figure 4c should label with WT and DKO.

10.   WT is not included Figure 5a but in figure 5b. How the heatmap gene sets are arranged?

11.   Figure 5e gene names are unable to read.

12.   Line 462, “KO MEFs have multiple defects in DNA damage recognition and repair pathways”. Which KO? It is concluded that “these findings indicated that KO MEFs were subject to greater baseline genotoxic stress than WT MEFs  and responded to further damage in an abnormal manner” (line 492) is not convincing because 1) the three KO transcripts are different (figure 5) and figure 5e shows the similarity between WT and MlxKO. 

13.   Figure 6b, it will be easier to see the response to dose curve using continuous curve lines than violin boxes for the demonstration of dose-dependent manner.

14.   Side by side KO comparison between the primary and immortalized cell lines will be helpful, especially for WT comparison considered as baseline.

15.   As in figure 7e, why avoid comparison of single KO vs DKO?

Reviewer 3 Report

The manuscript by Wang et al (cells-1967310) is a scholarly discourse on the interplay between MYC and MLX networks.  Because MLX heterodimerizes with bHLH proteins that may also heterodimerize with the MYC partner MAX, there is tension between these networks. The manuscript reports their analyses of target genes, the metabolic pathways as well as the physiological and pathological consequences of deregulation of these networks. These consequences heavily impact cell growth, senescence, apoptosis, DNA damage, etc. The manuscript is encyclopedic and is a repository of valuable information accrued from well-executed and well controlled experiments, that appear to me to have been appropriately analyzed statistically and computationally.

The only issue that I have with this magnum opus is that its sheer magnitude and density hamper the ability to extract a few clarifying “take-home” messages. Admittedly, though such take home messages are likely to represent over-simplifications.  Perhaps the authors could provide some hint as to their perceptions of the relative importance and novelties of their various results. Even without a punchy bottom line, this work will prove to be a valuable resource and archive of the interactions between these networks.  This data should be of great value to other investigators trying to understand the roles of the MYC and MLX networks in health and disease.

The manuscript is generally well-written and any specific comments I considered communicating were so minor that I decided not to include them.

Author Response

Comment: Perhaps the authors could provide some hint as to their perceptions of the relative importance and novelties of their various results. Even without a punchy bottom line, this work will prove to be a valuable resource and archive of the interactions between these networks. This data should be of great value to other investigators trying to understand the roles of the MYC and MLX networks in health and disease.

Response: In light of this comment and that of Comment 2 by Reviewer 1, we have re-worked our Abstract and portions of the Discussion to better emphasize some of the major “take home” points. Among these is that the reversal of primary MycKO MEF growth-arrest by either Mlx loss or SV40 T antigen immortalization involves inhibition of the p53 and/or Rb pathways.

Reviewer 4 Report

Good experimental design and results, please, reparagrap the abstract to connect ideas.  "The expression of DNA damage-related proteins was also abnormal. Immortalized KO MEFs remained proliferation-competent but demonstrated differential sensitivities to genotoxic agents.
Immortalized MycKO MEFs spontaneously developed tetraploidy that was Mlx-dependent. Dif- 30
ferent aspects of MEF aging, senescence and DNA damage responses are therefore differentially
regulated by the Myc and Mlx Networks."

Author Response

This Reviewer had no specific comments
In summary, we believe that we have adequately addressed the major comments raised by the four Reviewers. We thank them for their suggestions that have helped us to in strengthening the conclusions of the paper and its experimental underpinnings. We hope that our revised manuscript and this letter of response will help in making this paper acceptable for publication in Cells.

Round 2

Reviewer 1 Report

I thank the Authors for taking the time and making the effort of addressing my concerns. Regarding the Authors' answer to comment #4, I would kindly ask the Authors to include the data of the rescue (and its detailed description) in the final version of the article, either as a text or a suppl. figure.
